# High Frequency Protein-Rich Meal Service to Promote Protein Distribution to Stimulate Muscle Function in Preoperative Patients

**DOI:** 10.3390/nu13041232

**Published:** 2021-04-08

**Authors:** Vera IJmker-Hemink, Nicky Moolhuijzen, Geert Wanten, Manon van den Berg

**Affiliations:** 1Department of Gastroenterology and Hepatology, Dietetics and Intestinal Failure, Radboud University Medical Centre, 6525 GA Nijmegen, The Netherlands; Nicky.Moolhuijzen@radboudumc.nl; 2Department of Gastroenterology and Hepatology, Radboud University Medical Centre, 6525 GA Nijmegen, The Netherlands; Geert.Wanten@radboudumc.nl (G.W.); Manon.vandenBerg@radboudumc.nl (M.v.d.B.)

**Keywords:** protein distribution, muscle function, home-delivered meal services

## Abstract

Apart from meeting daily protein requirements, an even distribution of protein consumption is proposed instrumental to optimizing protein muscle synthesis and preserving muscle mass. We assessed whether a high frequency protein-rich meal service for three weeks contributes to an even daily protein distribution and a higher muscle function in pre-operative patients. This study was a post-hoc analysis of a randomized controlled trial (RCT) in 102 patients. The intervention comprised six protein-rich dishes per day. Daily protein distribution was evaluated by a three-day food diary and muscle function by handgrip strength before and after the intervention. Protein intake was significantly higher in the intervention group at the in-between meals in the morning (7 ± 2 grams (g) vs. 2 ± 3 g, *p* < 0.05) and afternoon (8 ± 3 g vs. 2 ± 3 g, *p* < 0.05). Participants who consumed 20 g protein for at least two meals had a significantly higher handgrip strength compared to participants who did not. A high frequency protein-rich meal service is an effective strategy to optimize an even protein distribution across meals throughout the day. Home-delivered meal services can be optimized by offering more protein-rich options such as dairy or protein supplementation at breakfast, lunch and prior to sleep for a better protein distribution.

## 1. Introduction

During hospitalization, muscle mass may decrease due to physical inactivity, underlying disease and/or malnutrition [1,2,3]. Low muscle mass has negative clinical implications for patients since it is associated with prolonged hospital stay, readmissions and higher mortality [4]. Protein stimulates muscle protein synthesis and thus, increasing protein intake holds promise as a strategy to beneficially modulate loss of muscle mass [2,5,6]. Because of the higher protein requirements for hospitalized patients a protein intake of 1.2–1.5 grams per kilogram body weight per day (g/kg/day) is recommended while maintaining energy balance [7]. Also, an even protein distribution across daily meals stimulates muscle protein synthesis more than a skewed protein intake towards one meal [8]. Muscle protein synthesis can be maximally stimulated by a distribution of 25–30 g of normal quality protein or 20 g of high quality protein per meal, three to four times per day [6,9,10,11,12]. High quality proteins/amino acids such as whey and leucine seem to have a higher anabolic response due to a high digestibility and essential amino acid content [13].

It is suggested that consuming at least one high-protein meal per day stimulates muscle protein synthesis among healthy adults with a protein intake of 0.8–1.3 g/kg/day [14]. In addition, achieving the threshold of 20 g at least twice a day might be beneficial for groups at high-risk of developing malnutrition, such as elderly or (chronically) ill patients [15,16]. Loss of muscle mass is especially seen in these groups due to sarcopenia, reduced food intake and a blunted anabolic response to protein ingestion [6,11,17]. Most literature on this topic, however, focused on frail elderly and evidence in the hospital setting is lacking [11,15,16,18]. The clinical patient population largely comprises vulnerable patients, underpinning the importance of adequate protein supplementation. Meeting protein requirements and distributing proteins evenly across meals might be crucial in order to preserve muscle mass and maintain functionality [6,11].

Recent literature shows that implementation of a high frequency hospital meal service substantially improves protein and energy intake in hospitalized patients within a short period of hospital stay [19]. Moreover, protein intake was improved at all meal times compared to a traditional three meals a day service but the minimal threshold of 20 g was only achieved at dinner [20]. Following the rising trend of out-of-hospital care, this high frequency protein-rich meal service was implemented at home for patients awaiting surgery resulting in a higher protein intake and a higher percentage of patients achieving their individual protein requirements (35% vs. 14%) [21]. Extending this high frequency hospital meal service to an out-of-hospital setting and thus prolonging the exposure period to adequate protein intake at home might be beneficial for the recovery after surgery. At this point, it remains unclear whether such a high frequency protein-rich meal service in the home setting contributes to an even distribution of protein intake during the day.

Therefore, a post-hoc analysis was performed on our previously collected data to assess whether offering a high frequency protein-rich meal service in the home setting for three weeks contributes to a more even protein distribution of meals over the day compared to usual care in pre-operative patients. The difference in handgrip strength as an indicator of muscle function was also assessed between participants who achieved the minimal threshold of 20 g proteins for at least two meals a day with participants who achieved this for less than two meals.

## 2. Materials and Methods

### 2.1. Study Population and Design

This was a post-hoc analysis of a randomized controlled trial (RCT), which was conducted at two hospitals in the Netherlands (ClinicalTrials.gov NCT03488511) [21]. The Medical Ethics Committee of the Radboudumc indicated that no formal approval was required for this study (2016–3043) (clinicaltrials.gov: NCT03488511). Participants were randomized in the usual care (UC) or intervention group by using block randomization, stratified for type of surgery and malnutrition risk using the Malnutrition Universal Screening Tool (MUST). Participants either received the high frequency protein-rich meal service in the pre-operative setting for three weeks or continued their habitual nutritional intake. Participants in both groups were visited at their home before and after these three weeks by trained nutritionists or dieticians to perform measurements.

The study population comprised Dutch-speaking pre-operative patients at the department of general surgery, orthopedics, gynecology or urology of Radboudumc and Maasziekenhuis Pantein. All participants were aged 18 years or older and lived within a 40 km radius of the Radboudumc or Veghel, the Netherlands. Patients were excluded from participation when they were on enteral or parenteral feeding, or when they suffered from renal insufficiency (MDRD-GFR < 60 mL/min and/or proteinuria). All participants gave written informed consent before participation.

### 2.2. Nutritional Intervention

Participants received the high frequency protein-rich meal service at home in the pre-operative setting for three weeks. This meal service consisted of six small, protein-rich dishes per day, including a morning snack (shakes: 5–10 g protein), lunch (salads and soups: 7–25 g protein), afternoon snack (5–19 g protein), dinner (hot meal and dessert: 16–39 g protein) developed by Radboudumc and a caterer (Appendix A) [21]. Data on energy and protein content of the meals was provided in the original publication [21]. Breakfast was not included, but participants received an information leaflet with protein-rich options for breakfast. Participants were not restricted to the FoodforCare meals only but were allowed to consume their own food products as well and received their individual protein requirements per day, information about how to meet these requirements and a tool to measure their daily protein intake relative to requirements. Participants randomized to the UC group were asked to continue their habitual diet and did not receive information leaflets. None of the participants were informed about the importance of achieving 20 g protein for at least two meals a day.

The distribution of protein intake over the day was evaluated based on a three-day food diary that was filled in by the participants both before and after the intervention period [22]. Type and amount of food and timing of intake were reported. During study visits, the food diaries were cross-checked by nutritionists. For each meal occasion (breakfast, during the morning, lunch, during the afternoon, dinner and during the evening), the amount of protein and kilocalories (kcal) consumed were calculated according to the Dutch Food Composition Table (NEVO, RIVM) with the software program Madows (PinkRoccade Healthcare, Apeldoorn, the Netherlands). The mean protein intake per meal occasion from the 3-day food diary was used for analysis. Total protein intake was expressed as the percentage of total grams of protein consumed per day relative to individual protein requirements with 1.2-g protein per kilogram of corrected body weight (g/kg BW) per day as the minimum requirement. To calculate protein requirements, body weight of patients with a body mass index (BMI) (calculated as weight in kilograms divided by height in meters squared) ≥30 was corrected to a body weight that corresponds with a BMI = 27.5 to avoid overconsumption. The body weight of patients with a BMI < 20, was corrected to a body weight-fitting BMI of 20 to avoid under consumption [23]. For patients with a BMI of 20–30, the actual body weight was used.

Total energy intake was expressed as the percentage of total kcal consumed per day relative to individual energy requirements using the World Health Organization (WHO) formula multiplied by 1.3 for illness and physical activity.

### 2.3. Secondary Outcome

Handgrip strength is a known indicator for decreased physical status and was therefore, used as a measurement of muscle function [24]. Handgrip strength was measured with the Jamar handgrip dynamometer (ADVYS, Waasmunster, Belgium). Participants were asked to sit upright with their elbows at a 90° angle. Handgrip strength was measured two times at both hands. The maximum score at each hand was defined as the participant’s handgrip strength [24,25].

### 2.4. Statistical Analysis

All continuous baseline characteristics were presented as mean ± Standard Deviation (SD) or median and interquartile range (IQR). Ordinal data were presented as frequencies and percentages. The differences in protein per meal occasion between the intervention and UC group were analyzed using independent samples *t*-tests. An Analysis of Covariance (ANCOVA) analysis was performed to analyze the difference in handgrip strength between participants achieving the minimal threshold of 20 g protein for at least two meals per day and participants who achieved this threshold for one or less meals per day. Age, gender, BMI and baseline handgrip strength were included as covariates in the ANCOVA analysis. For all analyses, a two-sided *p*-value < 0.05 was considered statistically significant. Data were analyzed using IBM SPSS Statistics V25.0 (IBM Corp, Armonk NY, USA).

## 3. Results

### 3.1. Demographics

A total of 102 participants (intervention: *n* = 52, UC: *n* = 50) were included for analysis. Baseline characteristics are shown in Table 1. Overall, baseline characteristics were comparable between the two groups.

### 3.2. Dietary Intake

Figure 1 shows the difference in median protein intake in grams per meal occasion for both the intervention group and UC group after the three week intervention period. In the intervention group, protein intake was significantly higher at the in-between meals in the morning (7 ± 2 g vs. 2 ± 3 g, *p* < 0.05) and afternoon (8 ± 3 g vs. 2 ± 3 g, *p* < 0.05). The highest protein intake was achieved at the three main meals in both groups. In both groups, dinner was the only meal occasion in which the minimal threshold of 20 g was achieved. Total energy intake relative to requirements after the intervention period was 106% ± 21 in the intervention group and 87% ± 25 in the UC group. The percentage of proteins consumed by the participants beside the meals of the intervention was 34%. A total of 34 out of the 102 participants (16 (31%) intervention group vs. 18 (36%) UC group) achieved the minimal threshold of 20 g protein intake for at least two meals per day after the three-week intervention period (Table 2). These patients also had a significantly higher protein intake relative to requirements (96 ± 35% vs. 80 ± 20%, *p* = 0.001) and energy intake relative to requirements (104 ± 26 vs. 93 ± 24%, *p* = 0.044).

### 3.3. Handgrip Strength

Participants who achieved the threshold for at least two meal occasions had significantly higher handgrip strength than participants who achieved this threshold for less than two meal occasions after correcting for age, gender, BMI and baseline handgrip strength (44 ± 15 vs. 31 ± 12 kg, *p* = 0.029).

## 4. Discussion

This post-hoc analysis shows that a high frequency protein-rich meal service significantly improves protein intake at the in-between meals in the morning and afternoon compared to usual care. The minimal threshold of 20 g protein was only achieved at dinner in both groups [9,10]. Besides, participants who achieved the minimal threshold of 20 g for at least two meals had a significantly higher muscle strength compared to participants who achieved this threshold for less than two meal occasions.

Consistent with our previous publication on hospitalized patients, protein intake remains the highest at the main meals in both groups after the three week intervention period [20]. Protein intake at the main meals was comparable between the groups but was higher in the intervention group at the in-between meals. This is an interesting finding since it is well-known that protein has a higher satiating effect than other macronutrients, which might lead to a reduction in overall food intake during the day [26,27,28]. Furthermore, it is suggested that consumption of protein-enriched products is compensated for by choosing low-protein products for the rest of the day [27]. Therefore, our findings imply that distributing protein in smaller protein rich meals during the day results in a higher overall daily protein and energy intake without compensating by consuming less protein at the main meals.

Multiple studies describe an improvement in overall protein intake upon implementation of a meal service in older adults but there is a lack of literature on the effect on protein distribution over the day [29,30,31,32,33,34]. However, several studies in institutionalized elderly show that an even distribution of proteins across three meals is more effective in increasing daily protein intake than a pulse distribution (majority of proteins in one meal) [18,35,36]. In that vein, there still seems to be room for improvement since the threshold of 20 g of proteins was only achieved at dinner (27 g) compared to 13 g at breakfast and 16 g at lunch. Other studies also report that breakfast is the main meal that contains the lowest amount of protein in the elderly [37,38]. Drinks for the main meals and breakfast were not included in our intervention but participants received an information leaflet with protein-rich options. Therefore, extending the service with more protein-rich options, such as dairy or protein supplementation, at breakfast and also lunch might improve the protein distribution [39]. Another strategy could be to optimize the protein provision in the evening. Multiple studies show that providing a dose of casein protein (20–40 g) prior to sleep stimulates muscle protein synthesis overnight [40,41]. Also, resistance-type exercise before this dose seems to further increase this beneficial effect of protein supplementation prior to sleep [42,43].

Our findings also indicate that achieving the threshold of 20 g for at least two meals a day is associated with higher handgrip strength as a surrogate marker for muscle function. These patients also achieved a higher daily protein intake relative to requirements which suggests that a more evenly protein distribution not only results in a higher daily protein intake but also in higher handgrip strength. This finding is consistent with a review concluding that adequate total protein intake is of great importance to promote muscle health and promoting an optimal protein distribution is a practical method to achieve this [14]. Also, two cohort studies show that free-living elderly with an even protein distribution (at least 20 g at the main meals) had higher lean mass and muscle strength compared to a skewed protein distribution [36,44]. These findings underpin the hypothesis that consuming at least one high-protein meal per day may support muscle function. It should be noted that handgrip strength was used as a surrogate marker for muscle function. Future research should include other measures of muscle function in addition to handgrip strength to confirm these findings. Furthermore, we expressed protein distribution in grams per meal occasion but there are other ways to show this variable such as, protein intake in grams/kg body weight or the percentage of protein per main meal relative to the total protein intake. There is no consensus on how protein distribution should be analyzed while this would ease the comparison between studies [45]. Jespersen&Agergaard suggest including body composition when protein intake is determined per meal. We added BMI to the ANCOVA model to correct for body composition. For future studies we recommend to assess fat free mass or muscle mass and either express protein intake as g/kg muscle mass per meal or add this as a covariate in the analysis.

A strength of our study is that our participants were not obliged to consume only the dishes from the high frequency protein-rich meal service but were free to consume other food products as well. Therefore, implementing this meal service into their diet was relatively easy for the participants. Besides focusing on protein intake, we also took energy intake into account as it is known that malnourished patients should also have an adequate energy intake to avoid protein catabolism [7,46]. On the other hand, a limitation is that we did not perform a power calculation for this study since this was a post-hoc analysis. However, power calculations in the original study were based on total protein intake which makes it likely that power calculations based on protein distribution are similar. Another limitation is that we did not advise participants in the intervention group to strive for an intake of 20 g of protein for at least two meals per day. It is likely that the number of participants achieving this threshold would be higher when this was part of the advice.

In conclusion, a high frequency protein-rich meal service is an effective strategy to promote an even distribution of protein across meals throughout the day. This even distribution contributes to meeting total protein and energy requirements per day and achieving the threshold of 20 g for at least two meals is associated with a higher muscle function in pre-operative patients. Home-delivered (high frequency) meal services can be further optimized by offering more protein-rich options such as dairy or protein supplementation at breakfast, lunch and prior to sleep.

## Figures and Tables

**Figure 1 nutrients-13-01232-f001:**
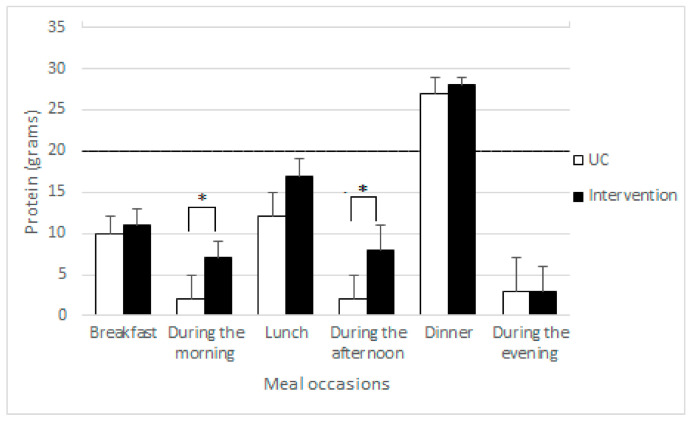
Difference in protein intake (mean ± SD) per meal occasion after the three week intervention period of the usual care and intervention group analyzed using independent samples *t*-tests. * Statistically significant between the groups, *p* < 0.05. Horizontal line at 20 g represents the minimal threshold that is suggested to be beneficial for muscle protein synthesis. Abbreviations used: UC: Usual Care; SD: Standard Deviation.

**Table 1 nutrients-13-01232-t001:** Baseline characteristics of 102 surgical patients.

Baseline Characteristics		UC Group ^1^ (*n* = 50)	Intervention Group (*n* = 52)
Gender, *n* (%)	Male	29 (58)	22 (42)
Age, years, mean ± SD ^1^		62 ± 13	63 ± 12
BMI^a^, kg/m^2^, mean ± SD ^1^		27 ± 4	28 ± 6
MUST ^1^, *n* (%)	0	41 (82)	44 (85)
	1	7 (14)	5 (10)
	≥2	2 (4)	3 (6)
Protein intake relative to requirements (%), mean ± SD ^1^		80 ± 25	77 ± 21
Energy intake relative to requirements (%), mean ± SD ^1^		85 ± 24	83 ± 23
Oncological disease, *n* (%)		12 (24)	12 (23)
Department, *n* (%)	General Surgery	15 (30)	16 (31)
	Orthopedics	11 (22)	11 (21)
	Urology & Gynecology	24 (48)	25 (48)
Handgrip strength, mean ± SD ^1^		35 ± 16	32 ± 12

^1^ Abbreviations used: UC: Usual Care; SD: Standard Deviation; BMI: Body Mass Index; MUST: Malnutrition Universal Screening Tool.

**Table 2 nutrients-13-01232-t002:** ANCOVA ^2^ analysis for the difference in handgrip strength between participants who achieved the threshold of 20 g at ≥2 meal occasions compared to <2 meal occasions.

	<2 Meal Occasions Threshold of 20 g ^1^	≥2 Meal Occasions Threshold of 20 g ^1^	*p*-Value
Handgrip strength (kg)	31 ± 12	44 ± 15	0.026
Protein intake relative to requirements (%)	80 ± 20	96 ± 35	0.001
Energy intake relative to requirements (%)	93 ± 24	104 ± 26	0.044

^1^ For handgrip strength: <2 meal occasions *n* = 67, ≥2 meal occasions *n* = 33 (intervention group *n* = 16 vs. UC group *n* = 18); ^2^ Abbreviations used: ANCOVA: Analysis of Covariance.

## Data Availability

The data that support the findings of this study are available upon request from the corresponding author.

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
