# Peer review of "High Frequency Protein-Rich Meal Service to Promote Protein Distribution to Stimulate Muscle Function in Preoperative Patients"

_nutrients, 2021, doi:10.3390/nu13041232_

Round 1

Reviewer 1 Report

The paper is a post-hoc analysis of a previous study. The original study was not blinded and a placebo effect is possible. Besides this design issue, the present analysis is incomplete and I suggest to perform more analysis on the data to support conclusions.

The critical issue is the relation between intake of protein and HGS. A simple alternative explanation could be that larger individuals consume more food but are also stronger as they have more muscle mass and strength. I believe the ANCOVA analysis should include a measure of this (maybe BMI) to control for this effect. 

Also the point about 20 gram protein or more per meal is really a little bit simple as a larger person will consume more per definition. If any, data need to be controlled for body weight, energy intake or whatever. It is very common to express data per kg BW, although I believe other calculations are possible like per kg ffm, lbm of muscle mass (which would make most sense).

Other comments

115: Repeat here how corrected body weight was calculated

125: I believe Jamar measurements is validated for 3 times measurement per hand

128: Why not present data are mean with 95% CI?

131: Why not logtransformation and then parametric testing?

135: add BMI or alike into the ANCOVA model

160 (figure 1): Display mean with 95% CI. Are these estimated values for ANCOVA?

247: explain CZ

Reviewer 2 Report

In the study by Ijmker-Hemink and colleagues the influence of a high frequency protein-rich meal service to pre-operative patients in relation to hand grip strength was explored. The study is of great relevance, however being a posthoc analysis with the attempt to study on evenness of protein distribution the methodologic approach and justification on the effect of the meal service is questionable.

Major comments

The focus seems to be switch through the manuscript, and it is not clear if the original purpose was to study the evenness in relation to the meal service or to perform a posthoc analysis on subjects eating 2 or more meals with more than 20 grams of protein pr meal. Reading the title, the reader expect the intervention group to have a more even distribution of dietary protein intake, which is not the case. They eat more protein in total. Please revise the title.

Furthermore, by not finding a difference in protein distribution (figure 1), it seems as if focus is switched to look at a subgroup of both the UC and intervention group that consume 2 or more meals per day with more than 20 grams of protein. If this is the evenness (across groups) and main purpose that you want to assess, then this should be clearly explained in the methods section.

More information is needed in order to clarify the protein distribution and protein amount of the meal and specific dishes and to justify the statements on evenness. What was the protein content of each dish? How was the compliance to the meals and how much protein did the participants consume beside the served meals?

The choice of a cut-off at 20 grams of protein per meal should be further explained and justified. It is stated that 20 gram could be enough if the protein is of high-quality, but what was the protein quality for both the UC and intervention group checked or controlled? It is indicated that the meals are served on basis of body weight – 1.2 gram protein/kg body weight/day. Why is the cut-off limit and evenness of protein distribution not based on body weight, as done and suggested by others? (PMID: 25056502, 30871197 and 33550490)

Minor comments

Line 38-40: Please rephrase to underline the digestibility and EAA content of whey protein and anabolic importance of leucine.

Line 45: “...due to sarcopenia..” is a broad statement as there is a numerous of factors contributing to sarcopenia and thereby the loss of muscle mass that is mentioned.

Line 62f: “..meal service in the home setting contributes to an improvement in even distribution..” What do you mean by improvement? Do you mean that the meal service contributes to even out the distribution of dietary protein?

Line 95: More information on the protein content of each dish of the meal service is needed.

Line 121: Handgrip strength has been shown to be related to muscle function, but it is still a direct measure of handgrip strength and not muscle function. Therefore, change here to handgrip

Round 2

Reviewer 1 Report

Excellent responses.

No further comments.